# Analysis of Ser/Thr Kinase HASPIN-Interacting Proteins in the Spermatids

**DOI:** 10.3390/ijms23169060

**Published:** 2022-08-13

**Authors:** Naoko Maeda, Junji Tsuchida, Yoshitake Nishimune, Hiromitsu Tanaka

**Affiliations:** 1Department of Science for Laboratory Animal Experimentation, Research Institute for Microbial Diseases, Osaka University, 3-1 Yamadaoka, Suita 565-0871, Osaka, Japan; 2Faculty of Pharmaceutical Sciences, Nagasaki International University, 2825-7 Huis Ten Bosch, Sasebo 859-3298, Nagasaki, Japan

**Keywords:** C1QBP, CENPJ, KPNA, RANBP9, MORN, Tcp10, Aip, Lims1, Pdlim5, Mad2l2

## Abstract

HASPIN is predominantly expressed in spermatids, and plays an important role in cell division in somatic and meiotic cells through histone H3 phosphorylation. The literature published to date has suggested that HASPIN may play multiple roles in cells. Here, 10 gene products from the mouse testis cDNA library that interact with HASPIN were isolated using the two-hybrid system. Among them, CENPJ/CPAP, KPNA6/importin alpha 6, and C1QBP/HABP1 were analyzed in detail for their interactions with HASPIN, with HASPIN phosphorylated C1QBP as the substrate. The results indicated that HASPIN is involved in spermatogenesis through the phosphorylation of C1QBP in spermatids, and also may be involved in the formation of centrosomes.

## 1. Introduction

The *HASPIN* gene was cloned from a subtraction cDNA library, constructed by subtracting the testis mRNA of a W/W^v^ mutant mouse from a wild-type testis cDNA library [1]. The gene encodes a serine/threonine kinase localized in the nucleus [2]. HASPIN is highly expressed in haploid spermatids, but studies have reported that it is also expressed at very low levels in various other tissues [3]. Moreover, studies have revealed that HASPIN is involved in regulating the functioning of chromosomes and spindles by phosphorylating histone H3 at Thr3 in mitotic cells [4,5]. TH2A is a germ-cell-specific H2A isoform. Studies have also suggested that HASPIN is involved in the phosphorylation of Thr127 of TH2A in condensed spermatids and in the mitotic early preimplantation embryos of mice [6].

HASPIN is widely conserved not only in animals but also in yeasts and plants [7,8]; thus, it presumably plays an important role in cell division [9]. HASPIN-gene-disrupted mice were produced in order to clarify the role of HASPIN [10]. Microscopy of the testes revealed some seminiferous tubules lacking germ cells but no other obvious phenotype. Recent advances in kinome projects have led to the isolation of HASPIN kinase inhibitors, and studies have reported that inhibitors of HASPIN kinase activity suppress the proliferation of cancer cells [11]. Abnormalities were not observed in individual HASPIN-disrupted mice, indicating that other molecules may compensate for the absent functioning of HASPIN in normal cells, but that HASPIN is crucial to the cell division of abnormal cells such as cancer cells.

In order to proceed with a detailed analysis of the function of HASPIN in normal cells, in contrast to abnormal cells such as cancer cells, we attempted to identify proteins that bind to HASPIN from the mouse testis cDNA library, using a two-hybrid system. Ten genes encoding binding proteins to HASPIN were identified, and detailed analyses of the relationship between HASPIN and each of these genes were achieved.

## 2. Results

### 2.1. Screening of Proteins That Interact with HASPIN in a Mouse Testis cDNA Library Using Two-Hybrid System

Using the full length of *Haspin* as bait, the mouse testis cDNA library was screened by the two-hybrid system. As a result of screening 5 × 10^5^ clones, sixteen positive clones with β-galactosidase activity were obtained. Ten genes were identified from their primary structural analyses (Table 1). Among the sixteen positive clones, five clones of *Cenpj*/*CPAP* genes, and two clones each of *Kpna6*/*importin alpha 6* and *Morn2/MORN repeat containing 2* were obtained. One clone of each gene of *C1qbp*/*HABP1*, *Ranbp9*/*RanBPM*, *Tcp10*/*t-complex protein 10*, *Aip*/*aryl-hydrocarbon receptor-interacting protein*, *Lims1*/*LIM and senescent cell antigen-like domain 1-like*, *Pdlim5*/*PDZ and LIM domain 5*, and *Mad2l2*/*MAD2 mitotic arrest deficient-like 2* was obtained.

### 2.2. Northern Blotting of Genes Interacting with HASPIN

As a result of gene expression analysis using total RNA, six genes, namely *Kpna6*, *Ranbp9*, *Cenpj*, *Tcp10*, *Morn2*, and *Lims1* were found to be predominantly expressed in the testis (Figure 1). Five genes, *Kpna6*, *Ranbp9*, *Cenpj*, *Tcp10*, and *Morn2*, showed markedly increased expression during spermiogenesis, appearing from three weeks of age. Three genes, *C1qbp, Aip,* and *Pdlim5* were expressed in the testis and brain. No signal was obtained for *Mad2l2* (data not shown). It is considered that *Mad2l2* expression was very low.

### 2.3. Binding Evaluation of HASPIN and Proteins Screened by Two-Hybrid System in Cultured Cells

In male germ cells, HASPIN is localized in the nuclei of spermatids. In somatic cells, HASPIN phosphorylated histone H3 at threonine-3 (H3T3ph), creating a docking site for the chromosomal passenger complex (CPC). To confirm the specific binding between the products of genes screened by the two-hybrid system and HASPIN, the recombinant *Egfp*-*Haspin* was co-transfected in HEK 293 cells transfected with *Rfp*-*Cenpj-Flag*, *Rfp*- *Kpna6-Flag*, or *C1qbp*-*Flag*, respectively, and immunoprecipitation was performed from the lysate of each culture cell. As a result, a signal was detected with the same molecular size as the anti-FLAG antibody, in each immunoprecipitant with the anti-HASPIN antibody (Figure 2). These results indicated that CENPJ, KPNA6, or C1QBP obtained by the two-hybrid system bound to HASPIN in culture cells.

### 2.4. Expression of KPNA6/Importin α6 in the Testis

*Kpna6* transcribed alternative transcription initiation sites of E1 or E1A. In the 5′UTR region, transcription from E1 was ubiquitous, and E1A was specifically transcribed in spermatogenic cells (Figure 3) [12]. Translations from both mRNAs translated the same protein from similar open reading frames. Six homologue genes were reported in *Kpna* genes in the mouse, and the *Kpna6* is an essential gene for spermatogenesis [12]. KPNA6 and KPNA1 show high homology of 81% [13]. To confirm the specificity of our anti-KPNA6 antibody to the N-terminal region, western blotting was performed using the anti-KPNA6 antibody. Western blotting showed that the anti-KPNA6 antibody reacted not with KPNA1, but with KPNA6 (Figure 4A). The expression of KPNA6 was examined by western blotting using the antibody of KPNA6. The results showed that the KPNA6 protein increased with the appearance of haploid germ cells (Figure 4B). Western blotting of cell fractionation revealed signals of KPNA6, and detected cytoplasmic and nuclear fractionation (Figure 4B). In the immunostaining of the testis, KPNA6 was localized in round spermatids in a punctate manner near the nucleus in a cup shape, so as to surround the nucleus with the elongating spermatids (Figure 5). Faint signals were observed in spermatocytes or spermatogonia, or in Sertoli cells or Leydig cells, when using our antibody at the N-terminus. KPNA6 was also present in sperm, and the signal in sperm was matched by double staining with peanut agglutinin (PNA), demonstrating that KPNA6 was localized in acrosomes (Figure 5). KNAP6 was not observed in sperm that had completed the acrosome reaction (Appendix A).

### 2.5. Expression of CENPJ/CPAP in the Testis

CENPJ/CPAP belongs to the centromere protein family. It has been identified as a protein that interacts with centrosomal P4.1–135 protein in the centrosome. HASPIN plays an important role in the chromosomal passenger complex (CPC) in the centromere [5]. KPNA6 [16], RANBP9 [17], and CENPJ [18] are involved in the polymerization of microtubules. Our results, described above, imply a bond between HASPIN and these molecules. Next, the anti-CENPJ antibody was prepared and its expression in the testis was examined. As a result of western blotting, CENPJ was found to be intensely expressed in the 21-day-old testis, in which haploid sperm cells appeared (Figure 6). Its expression was detected in the cytoplasmic fraction. As a result of immunostaining, CENPJ was observed as punctate regions from spermatocytes, round spermatocytes near the nucleus, and in flagella in elongated spermatids (Figure 7). Chromatoid bodies are known as spermatid-specific structures that lie in spermatocytes and in round spermatids. Immunocostaining was performed with the anti-CENPJ antibody and the monoclonal antibody TRA54 [19], which specifically reacts with the chromatoid body. Both signals’ localization was matched in round spermatids, indicating that CENPJ is localized to the chromatoid body in sperm cells (Figure 7E).

### 2.6. Expression of C1QBP/HABP1 in the Testis

C1QBP was originally identified as a protein having specific affinity to hyaluronan (HA) [20]. C1QBP is expressed in various organs and has been reported to be involved in various types of cell bioactivity, such as RNA splicing and transcription [21,22,23,24,25,26]. It has been suggested that C1QBP plays an important role in sperm motility and binding to eggs during fertilization [27,28,29]. C1QBP has been shown to be phosphorylated in sperm that have acquired motility [30]. It has also been shown that C1QBP is activated by phosphorylation and is involved in cancer progression. As a result of western blotting using the anti-C1QBP antibody, a signal of 50 kDa specifically in the testis and 32 kDa in the other organs was observed (Figure 8). The signals of 32 and 50 kDa were observed in the infant testis, with cell fractionation observed at 50 kDa in the cytoplasm and nucleus, and 32 kDa in the cytoplasm only. These results are consistent with previous reports indicating that when the full-length protein of C1QBP is expressed, 73 amino acids at the N-terminus are processed, but the full-length protein that is not processed in the testis is specifically expressed [31]. It has been reported that C1QBP is localized in the nuclear acrosome and mitochondria [32]. As a result of immunostaining of the testis, C1QBP was found to be intensely localized in the acrosome, because the staining was consistent with FITC-PNA (Figure 9).

### 2.7. Phosphorylation Target Protein of HASPIN

HASPIN is a ser/thr kinase. A kinase assay using an immunoprecipitate examined whether KPNA6, CENPJ, and C1QBP were phosphorylated with HASPIN. As a result, the signal of autophosphorylated HASPIN was observed in all immunoprecipitates, while a signal of phosphorylation was obtained only in C1QBP (Figure 10). Moreover, no phosphorylation signal of HASPIN was obtained when using a mutant HASPIN in which a part of the kinase domain was deleted (Figure 10). These results indicated that the kinase activity of HASPIN directly or indirectly phosphorylated C1QBP.

## 3. Discussion

In the testis, germ cells proceed in an orderly manner from the outside of the seminiferous tubule to the lumen [33]. Spermatogonia, which are germline stem cells, produce differentiated cells while undergoing mitosis. The differentiated spermatocytes undergo meiosis and then proceed to cell differentiation into haploid spermatids. Round spermatids proceed to nuclear condensation, flagella formation, and acrosome formation, and differentiate into elongated sperm cells. HASPIN was cloned as a ser/thr kinase that is exclusively expressed in haploid sperm cells [2]. Elsewhere, HASPIN was expressed in small amounts in somatic cells [3] and has been found to play a role in chromosome distribution [4]. However, no critical phenotypes were observed in the analyses of HASPIN-gene-disrupted mice [10]. Thus far, autophosphorylation, Histone H3, and Histone H2t have been reported as substrates for the phosphorylation of HASPIN, but the role of HASPIN in spermatogenesis is unknown [2,4,6]. Here, using a testis cDNA library, a two-hybrid system was used to analyze proteins that bind to HASPIN. As a result, ten types of genes were cloned, and it was clarified that, among these, C1QBP is a phosphorylation substrate of HASPIN.

The analysis revealed the binding domains of the molecules that interacted with HASPIN, namely the glycine repeat, the morn (membrane occupation and recognition nexus) motif [34], the LIM domain [35], the TPR (tetratrico peptide) repeat [36], and the HORMA (Hop1p, Rev7p, and MAD2) domain [37]. The glycine repeat was recently reported as one of the characteristic domains of CENPJ. It has been reported that SATA5 binds to the glycine repeat in CENPJ by the yeast two-hybrid method [38]; however, its function is still poorly understood. Since the interaction between HASPIN and CENPJ was performed via this domain, the glycine repeat was considered to be a domain that functions for protein–protein interaction. Meanwhile, the morn motif has only recently been reported as possessed by Junctophilins, a complex protein that connects cell membranes and the endoplasmic reticulum [34], and functional analysis has seen little progress, in contrast to the glycine repeat. Thus, the functional relationship between HASPIN and MORN2 is unknown.

The LIM domain and TPR repeat are each involved in protein interactions and are found in many proteins. Among the group of proteins with the TPR repeat, the basic subunits (cdc27, cdc16, cdc23) of the anaphase promoting complex/cyclosome (APC/C) complex involved in the cell cycle have been reported [39]. APC/C, a type of ubiquitin ligase, is a protein that acts on sister chromosome segregation and the completion of the M phase. It is highly active from the late M phase to the G1 phase and controls cell-cycle-related molecules such as cyclin B. Because HASPIN functions in cell division, this result suggested an important relationship between the LIM domain, TPR repeat and HASPIN. Additionally, the MAD2B homolog with the HORMA domain was identified as a molecule that interacts with HASPIN [40]. This MAD2B homolog is thought to be involved in the cell cycle and DNA repair. In addition, the MAD protein is an activity inhibitor of APC/C, and it has been suggested that the phosphorylation of APC/C and MAD plays an important role in complex formation and activation control [41,42,43]. HASPIN regulates the activity of the APC/C complex by promoting the phosphorylation of APC/C and MAD proteins, when interacting with APC/C and MAD proteins via the TPR repeat and HORMA domain. The protein complex including HASPIN may be associated with the cell cycle.

Previously, it has been reported that KPNA6 is expressed in the cytoplasm of germ cells using the anti-C-terminal of the KPNA6 antibody [12]. In the N-terminal antibody that we prepared, differences of intracellular localization were observed by immunostaining, compared to the previous report of the C-terminal antibody by Liu et al. [12]. The signals of KPNA6 localized in the cytoplasm and nucleus in the testis, using the anti-N-terminal of the KPNA6 antibody. Our observation of KPNA6 was consistent with the localization of the nuclear–cytoplasmic transport protein, because it is considered that KPNA6 plays the role of transporting the protein into the nucleus. The other results of immunostaining for the anti-N-terminal of the KPNA6 antibody revealed that KPNA6 was intensely localized in the acrosome. Since KPNA6 is involved in the protein transport mechanism [44,45], it is possible that it plays an important role in protein transport to the acrosome. The acrosome is a type of vesicle containing proteins such as various hydrolases required for fertilization, and it is generally considered that the transport of proteins into the acrosome is carried out by the molecules involved in Golgi transport [46,47,48]. In germ cells, a lipid bilayer structure called the annulate lamellae, which has a structure similar to the nuclear pore, can be observed in the cytoplasm, and it has been reported that it forms vesicles and is taken up by the endoplasmic reticulum [49]. However, its physiology is not yet well understood [50]. It has been suggested that proteins with nuclear localization signals are carried to the annulate lamellae by the importin α/β transport mechanism [51], so the KPNA6/importin β complex may transfer binding molecules such as HASPIN to acrosomes through the annulate lamellae. It was revealed that KPNA6 is also present in acrosomes in mature sperm. It has been reported that cytoskeletal proteins including tubulin are significantly changed during the acrosome reaction process that occurs during fertilization [52,53]. From this, it is possible to infer that KPNA6 is involved in microtubule polymerization and regulates the acrosome reaction. Another function of importin α has been reported to involve microtubule polymerization in the M phase [16]. The CPC dynamically localized to different subcellular locations to regulate key mitotic phases, such as the correction of kinetochore–microtubule attachment errors, activation of the mitotic spindle assembly checkpoint, and assembly and maintenance of the outer kinetochore [54]. HASPIN plays a role in coordinating the functioning of the chromosomal passenger complex (CPC) in cell division [5]. In this study, CENPJ and RANBP9 were identified as proteins that bind to HASPIN, which in addition to KPNA6 are genes involved in protein complexes in the regulation of microtubules.

CENPJ has been identified as a protein that binds to the centrosome P4.1 and has been suggested to be involved in microtubule polymerization, but its function is not yet well understood. CENPJ was found to be present in the chromatoid body in spermatocytes. RANBP9, which was isolated as a molecule that binds to HASPIN, was also localized in the centrosome and may be involved in microtubule polymerization [17]; it is localized in the chromatoid body [14]. From these results, it was considered that HASPIN may play a role in protein complexes with CENPJ and RANBP9 in the chromatoid body. The chromatoid body is constituted by ribonucleoprotein and RNA. The mRNAs transcribed in spermatocytes are untranslated until they reach spermatid cells in the chromatoid body [55,56]. Although the presence of microtubules in the chromatoid body has not been observed, there is a close relationship between the formation and placement of microtubules and chromatoid bodies, because a microtubule inhibitor caused formation and distribution abnormalities of the chromatoid body [57]. From this, it was considered that CENPJ and HASPIN were involved in the formation and maintenance of the chromatoid body through microtubule polymerization. TCP10, which is highly homologous to the C-terminal sequence of CENPJ, is present on chromosome 17 in mice, and mice with reverse mutations in this region experienced sperm fertility distortion in heterozygous and infertility in homozygous cases [58,59,60]. Here, TCP10 was isolated and was revealed to bind to HASPIN in cultured cells (data not shown). The function and localization of TCP10 in the testis are not yet known. It would be interesting to determine the relationships among HASPIN, CENPJ, and TCP10. KPNA6 and CENPJ did not target phosphorylation substrates of HASPIN. Furthermore, since CENPJ and TCP10 are localized in sperm-cell-specific structures, HASPIN is thought to be involved in sperm cell morphogenesis through interaction with these molecules.

C1QBP has been identified as a molecule that binds to hyaluronic acid [20], and its physiological functions are diverse, suggesting that it is involved in cell proliferation, the cell cycle, infiltration ability, the transcription of mRNA, and splicing [21,22,23,24,25,26]. It was suggested that C1QBP bound to HASPIN and was potentially a direct phosphorylation substrate of HASPIN. However, the other phosphorylation signals of high molecular weight were observed in the kinase assay, suggesting that other kinases might be involved in the phosphorylation of C1QBP. It has been suggested that the phosphorylation of C1QBP activates the phospholipase C-inositol trisphosphate pathway [23,61]. Inositol phosphate pathways in sperm are thought to be responsible for signaling important intracellular calcium influx during the acrosome reaction [62]. It has been reported that inositol trisphosphate receptor phospholipase C (PLC) [63] and PLCδ4 were present in acrosomes, and that PLCδ4 gene-deficient mice were affected by male infertility [64]. From the above, it is considered that HASPIN activates the inositol phosphate pathway by phosphorylating C1QBP, and is involved in the acrosome reaction. In somatic cells, it has been shown that HASPIN functions in cancer cells and that HASPIN inhibitors suppress the growth of cancer cells [11,65]. Overexpression of C1QBP in cancer cells has been reported [66]. The phosphorylation-induced activation of C1QBP by HASPIN in cancer cells may be involved in the growth of cancer cells. Otherwise, C1QBP may be involved in chromatin aggregation; it has been suggested that it binds to the lamin B receptor (LBR), a component of the nuclear matrix that binds to the heterochromatin protein, HP1 [67]. In haploid sperm cells, histones are replaced by protamines and the nucleus is condensed. Since it has been clarified that HASPIN is localized in the nucleus and binds to DNA in vitro [2], it is also possible that HASPIN is involved in nuclear condensation in haploid germ cells by interaction with C1QBP.

Molecules that bind to HASPIN include proteins in spindle fiber checkpoints and in microtubule polymerization. From this, it was inferred that HASPIN regulated the cell cycle, especially for proteins from the end of the M phase to the G1 phase, and was involved in signal transduction and cytoskeleton organization associated with the acrosome reaction. In addition, since these genes are targets for the suppression of the growth of cancer cells, this is considered to be an important finding in elucidating the mechanism of cancer suppression. Further analysis of the relationships between molecules that interact with HASPIN will deepen our understanding of spermatogenesis, cell division, and cancer cell proliferation.

## 4. Materials and Methods

### 4.1. Animals

C57BL/6J mice were purchased from Japan SLC (Shizuoka, Japan) and sacrificed by cervical dislocation immediately prior to experiments. All animal experiments conformed to the Guide for the Care and Use of Laboratory Animals and were approved by the Institutional Committee of Laboratory Animal Experimentation and Research Ethics Committee of Nagasaki International University (approval number 128). This article does not contain any experiments with human subjects performed by any of the authors. Mice were maintained under specific pathogen-free conditions in the animal experimentation facility at Nagasaki International University, with temperature and lighting controlled throughout the experimental period. Mice were provided with food and water *ad libitum*.

### 4.2. Yeast Two-Hybrid System

The Matchmaker two-hybrid system 2 was used according to the protocol recommended by the manufacturer (Clontech Laboratories, Palo Alto, CA, USA). The full-length *Haspin* cDNA was constructed into a pAS2-1 vector (Clontech Laboratories, Palo Alto, CA, USA) and was transformed into competent yeast Y190 cells. The yeast expressed the recombinant protein of the GAL4-DNA binding domain fusion, and HASPIN was obtained as the tryptophan non-requiring strain. The mouse testis cDNA library was prepared in pACT2, and a vector expressed the fusion protein of the GAL4 transcription-activation domain (Clontech Laboratories, Palo Alto, CA, USA). The pACT2 included in the cDNA library was transformed into tryptophan non-requiring yeast and screened for genes expressing proteins interacting with HASPIN. Positive clones were obtained as tryptophan, leucine, and histidine non-requiring, and were found to express LacZ.

### 4.3. Northern Blots and DNA Sequencing

Total RNA was isolated from various tissues of mice (C57BL/6 strain), with TRIzol (Invitrogen, CA). Germ and other somatic cells of the testes were prepared as described in our previous report [68]. Total RNA was extracted according to the manufacturer’s recommendations, quantified by optical density measurement. A quantity of 20 μg of mRNA containing 2.2 M formaldehyde was separated on a 1.0% agarose gel with 0.66 M formaldehyde and transferred to a nitrocellulose filter. Hybridization was performed with ^32^P-labeled cDNA prepared with the BcaBest random primer kit (TAKARA, Shiga, Japan). Signals of the bands were detected by an image analyzer (Fuji Film, Tokyo, Japan). Dideoxy-chain termination sequencing reactions were performed with fluorescent dye-labeled primers and thermal cycle sequencing kits purchased from Applied Biosystems (Applied Biosystems, Waltham, MA, USA). The reaction products were analyzed by GNESCAN-373A (Applied Biosystems, Waltham, MA, USA).

### 4.4. RT-PCR

RNA from mouse organs was extracted as per the protocol of the RNAqueous kit (Life Technologies Tokyo, Japan). One μg of RNA was reverse-transcribed using the random primer (TAKARA) and Superscript II enzyme (Invitrogen) according to the manufacturer’s recommendations. PCR analysis of samples was performed using EX Taq HS polymerase (TAKARA). The 5′ UTR1 of *Kpna6* was amplified using the upper primer 1 (5′-GCTACCGCTGCGGCCGCCGCC-3′) and the reverse primer (5′-TGGAGCAATTCAAGACATAA-3′) in the open reading frame of *Kpna6*. The 5′ UTR2 of *Kpna6* was amplified using the upper primer 1 (5′-AGGACAGCCTCTGGGACAGCAA-3′) and the above reverse primer. The following PCR conditions were used: 30 cycles of denaturation at 94 °C for 30 s, annealing at 60 °C for 30 s, and extension at 72 °C for 40 s. PCR products were electrophoresed in 1% agarose and detected with ethidium bromide.

### 4.5. Western Blotting

First, 50 μg of tissue protein or 10 μg of fractionated protein was separated by SDS-PAGE, then electroblotted onto PVDF membranes. Nuclear and cytoplasmic fractions of the testicular germ cells were prepared as described in our previous report [14]. After blocking with 4% skimmed milk in TBS-T (50 mM Tris-HCl pH 7.4, 138 mM NaCl, 2.7 mM KCl, 0.1% Tween 20), the membranes were reacted with primary antibodies overnight at 4 °C, then secondary antibodies for 90 min at room temperature. The antigen–antibody complexes were detected using ECL Prime (GE Healthcare Japan, Tokyo, Japan). The antibodies used in the experiment are shown in Table 2.

### 4.6. Antibody Preparation

To obtain anti-KPNA6 or CENPJ antibodies, rabbits were immunized more than eight times with each synthetic peptide as an antigen (KPNA6: NVELINEEAAMFDSLC, CENPJ: CQQKKQEQLKRQQLEQLQ) in Freund’s adjuvant, respectively. Each IgG was purified from each serum using protein A-sepharose 4B Fast Flow (Amersham Bioscience, Amersham, UK).

### 4.7. Expression Vectors

KPNA6 with RFP fused to the N-terminus and FLAG epitope tag fused to the C-terminus were constructed into an expression vector, pDs-RedII-C1, regulated by a cytomegalovirus (CMV) promoter. *Kpna6* cDNA was amplified by using the primer set of 5′-EcoRI-catggcaagcccagggaagg-3′ and 5′-BamHI-tcattacttgtcgtcatggtcttgtagtcttgtagctggaagccctcca-3′ (TER taa to caa). *Kpna6* cDNA was amplified by using the primer set of 5′-EcoRI-catgtccacaccaggaaa-3′ and 5′-BamHI-tcattacttgtcgtcatcgtctttgtagtctccaagctggaaaccttccat -3′ (TER tga to gga). Each DNA fragment was inserted into pDs-RedII-C1 using EcoRI and BamHI restriction enzymes. The CENPJ expression vector was constructed using the primer set of 5′-SalI-atgttcctgatgccaacct-3′ and BamHI-tcattacttgtcgtcatcgtctttgtagtctcccatttctgtgtccattag-3′ (TER tga-gga), by the same process as described above for Kpna6. The C1QBP expression vector was constructed into NheI-BamHI sites in pDs-RedII-C1 without RFP fusion on the 5′-terminal, by using the primer set of 5′-NheI-Kozac-atgctccctcgctgcgttgc-3′ and 5′-BamHI-tcattacttgtcgtcatcgtctttgtagtcctgctggttcttgacaaagc-3′ (TER tag to cag). The nucleotide sequences of the cDNA were confirmed by thermal cycle sequencing. The previously reported expression vectors of pCNXII-*Egfp*-*Haspin* and pCNXII-*Egfp*-mutant *Haspin* were used [2].

### 4.8. Transfection

Transfection was performed according to the manufacturer’s recommendations using Lipofectamine (Invitorogen, Waltham, MA, USA). Here, 5 μg of each expression vector including *Kpna6*, *Cenpj,* or *C1qbp*, and *Egfp-Haspin* was co-transfected to HEK 293 cells. The cells were harvested after culturing for 48 h.

### 4.9. Immunostaining

Testes were immersed in OTC embedding compound (Tissue-Tek, Sakura, Tokyo, Japan) and were frozen at −20 °C. Sections (10 μm thick) were prepared using a cryomicrotome (HM 500 OM; Microm, Walldorf, Germany). Mature sperm were removed from the cauda epididymis and spotted onto glass slides (Matsunami Glass, Osaka, Japan). Samples were fixed with 80% methanol (for detection of KPNA6 and C1QBP) or 4% PFA (for detection of CENPJ) at 4 °C for 20 min before being permeabilized in PBS containing 0.01% Triton X-100 for 10 min. Each sample was treated with a blocking kit (Vector Labs, Newark, CA, USA) and blocking solution (Nacalai Tesque, Kyoto, Japan). The samples were incubated at 4 °C for 16 h with each antibody. After washing with TBS-T, the sections were incubated at room temperature for 90 min with biotinylated anti-rabbit Igs antibody (Amersham Bioscience). After washing with TBS-T, signals were observed using streptavidin/avidin-conjugated Alexa Fluor 568 (Amersham Bioscience). The sections were counterstained with 4′,6-diamidino-2-phenylindole (DAPI; Nacalai Tesque) and/or 20 µg/mL FITC-conjugated peanut aglutinin (FITC-PNA) (Sigma, St. Louis, MO, USA), washed, and examined under a fluorescence microscope. TRA54 and 98 were used to identify the chromatoid body and the nuclei of germ cells, respectively [17,69].

### 4.10. Immunoprecipitation

Tissue and cells were washed three times with PBS and lysed with RIPA (50 mM Tris-HCl pH 7.5, 150 mM NaCl, 1.0% NP-40, 0.5% sodium deoxycholate, 0.1% SDS). After homogenization with supersonication and centrifugation at 13,000× *g* for 5 min at 4 °C, the supernatant was gently mixed with Sepharose 4B at 4 °C for 1 h to eliminate binding with non-specific beads. The supernatant was treated with Protein A Sepharose 4B Fast Flow (Amersham Bioscience, Amersham, UK), according to the manufacturer’s instructions. The contents of each recovered fraction were analyzed by western blotting.

### 4.11. In Vitro Kinase Assay

Each immunoprecipitate was washed 3 times with the reaction solution (40 mM HEPES pH7.4, 10 mM MgCl_2_, 3 mM MnCl_2_, 5 mM CaCl_2_, 150 mM NaCl). Then, 10 μCi r^32^P-ATP was added to the immunoprecipitate, adjusted to 40 μL. The reaction was carried out at 37 °C for 10 min. After the reaction, a sample buffer for SDS-PAGE was added and centrifuged, and then we applied SDS-PAGE. The signals were observed using an image analyzer (Fuji Film) [2].

## Figures and Tables

**Figure 1 ijms-23-09060-f001:**
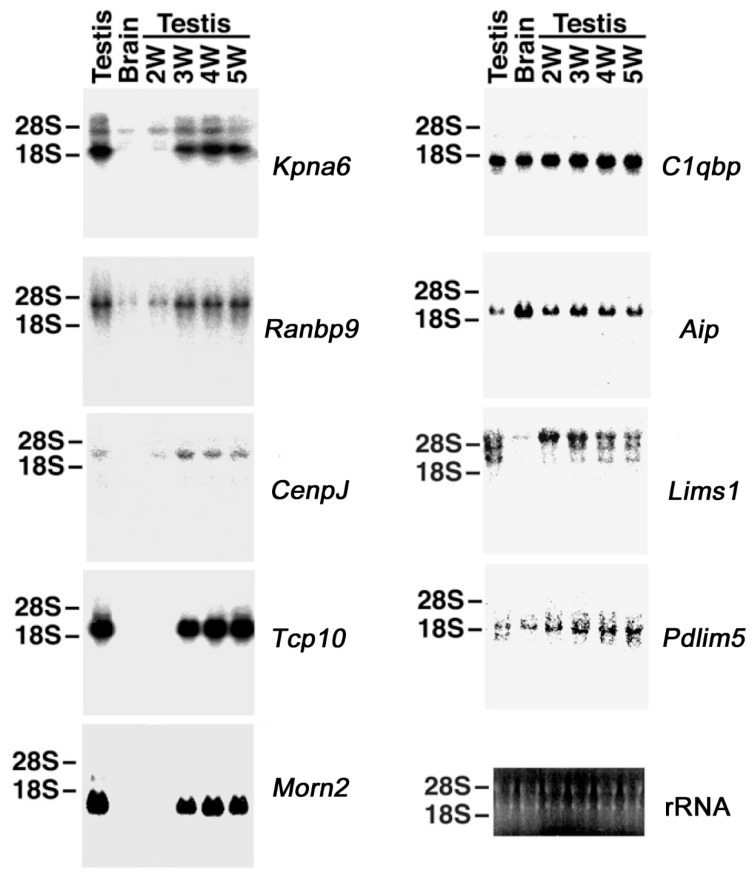
Expression of genes that interact with HASPIN, obtained through screening of the mouse testis cDNA library. Total RNA samples (20 μg) from adult testes, brains, and testes of two-, three-, four-, and five-week-old mice were subjected to northern blotting, using full-length cDNA as a probe. The position of ribosomal RNA is shown on the left side, and the name of each gene is shown on the right side. The lower-right panel shows the ethidium bromide staining after agarose electrophoresis.

**Figure 2 ijms-23-09060-f002:**
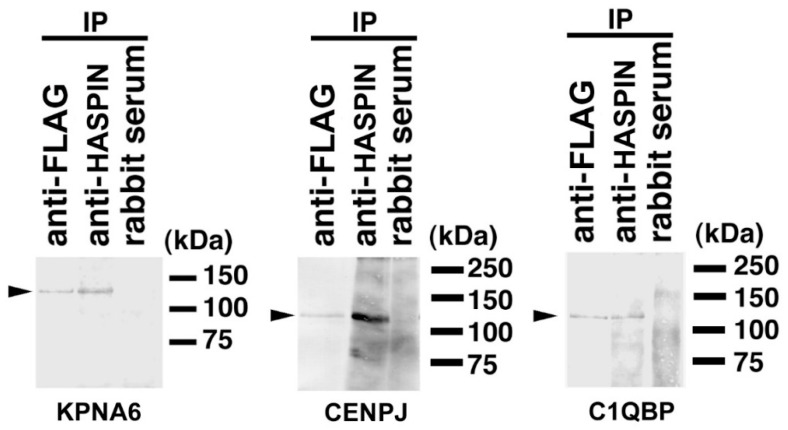
In vitro binding assay in HEK 293 cultured cells. The recombinant *Egfp*-*Haspin* expression vector was co-transfected in HEK 293 cells with the *Rfp*- *Kpna6-Flag*, *Rfp*-*Cenpj-Flag*, or *C1qbp*-*Flag* expression vector, respectively. After 48 h, immunoprecipitation was carried out from the lysate of each culture cell. The antibodies used for immunoprecipitation are shown above the panel. Genes co-expressed with *Haspin* are shown on the line at the bottom. The signals of HASPIN recombinant protein detected in each gene co-transfected the HEK 293 cells. No signals were detected in any of the immunoprecipitates, with rabbit serum used as a control. Molecular weights are shown to the right of each panel. Arrowheads indicate signals of EGFP-HASPIN.

**Figure 3 ijms-23-09060-f003:**
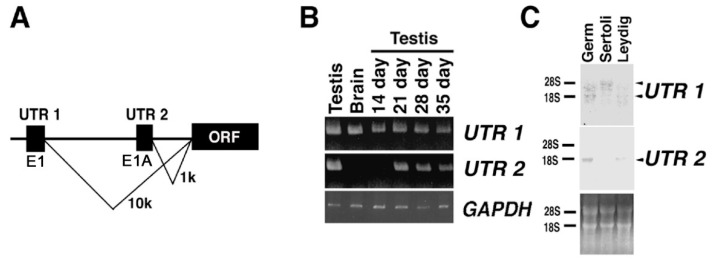
Expression analyses of *the Kpna6* gene. The transcription start sites of KPNA6 are schematically represented. (**A**) Alternative exons of UTR1 and UTR2 are reported [12]. RT-PCR was carried out by each UTR-specific primer pair using 609–628 (NCBI CCDC database: CCDS18701.1) in coding sequence as a reverse primer and UTR1-specific primer or UTR2-specific primer. (**B**) Northern blotting was carried out using testicular cell fraction. (**C**) The signal was specifically detected in the germ cell fraction using the UTR2-specific probe in each total RNA of germ, Sertoli, and Leydig cell fractions. The signals were detected in whole fractions using the UTR1-specific probe. The position of ribosomal RNA is shown on the left side, and each UTR signal is shown on the right side. The lower panel shows the ethidium bromide staining after agarose electrophoresis.

**Figure 4 ijms-23-09060-f004:**
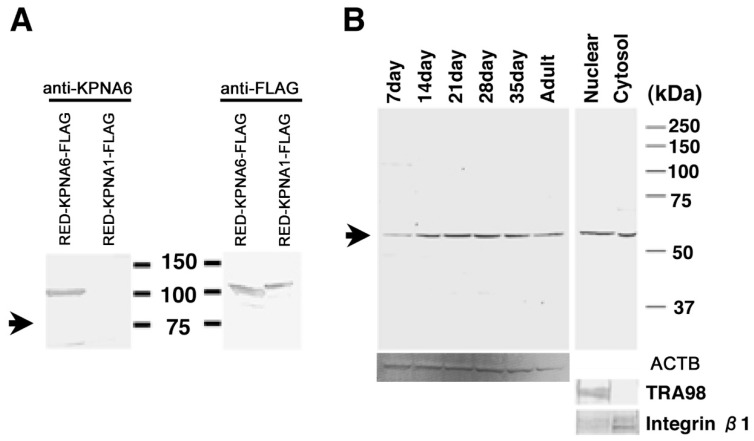
Western blotting of KPNA6 in testes. The specificity of the KPNA6 antibody was investigated using the extracts of HEK293 cells into which *Rfp*- *Kpna6-Flag, Rfp*- *Kpna1-Flag* expression vectors were transfected. The panels on the right and left in (**A**) show signals of the anti-KPNA6 or the anti-FLAG antibody, respectively. The anti-KPNA6 antibody did not react with KPNA1. (**B**) The expression of KPNA6 in testes was identified using 50 μg lysates of testes of one-, two-, three-, four-, five-, and ten-week-old mice, and 30 μg lysates of nuclear and cytosol fractions. Molecular weights are shown to the right of each panel. Arrowheads indicate signals of KPNA6. The antibodies of ACTB, TRA98 and anti-Integrin β1 were used as markers of nuclear or cytosol fractions in testicular cells, respectively [14,15].

**Figure 5 ijms-23-09060-f005:**
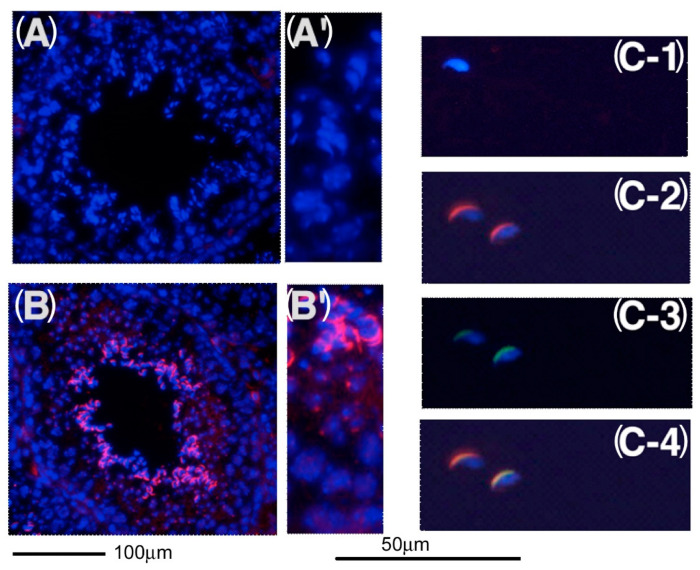
Localization of KPNA6 in testicular cells. Sections of adult testes were stained with (**A**,**A**’) pre-immune serum, and (**B**,**B**’) anti-KPNA6 antibodies. Sperm were stained with (**C-1**) pre-immune serum, (**C-2**) anti-KPNA6 antibodies, and (**C-3**) FITC-PNA. Nuclei were stained with DAPI. (**C-4**) The staining with anti-KPNA6 antibodies and FITC-PNA was merged.

**Figure 6 ijms-23-09060-f006:**
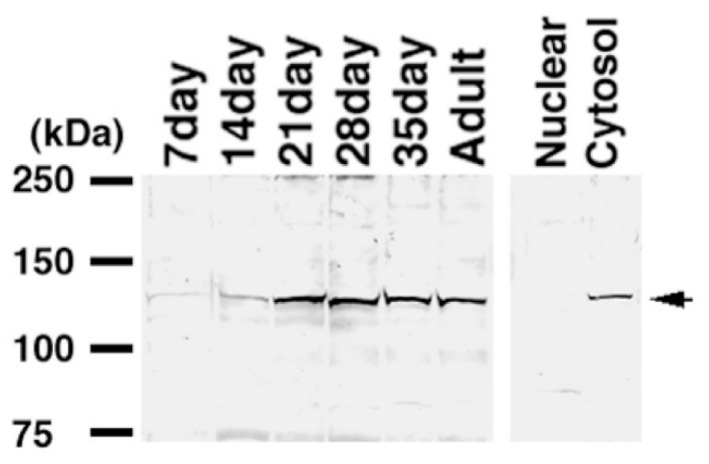
Western blotting of CENPJ in testes. The expression of CENPJ in testes was identified with the anti-CENPJ antibody using 50 μg lysates of testes of one-, two-, three-, four-, five-, and ten-week-old mice, and 30 μg lysates of nuclear and cytosol fractions. The arrows indicate signals of CENPJ. Molecular weights are shown on the left side. The protein samples in Figure 6 were adjusted at the same time as Figure 4, and the filter was adjusted in the same way.

**Figure 7 ijms-23-09060-f007:**
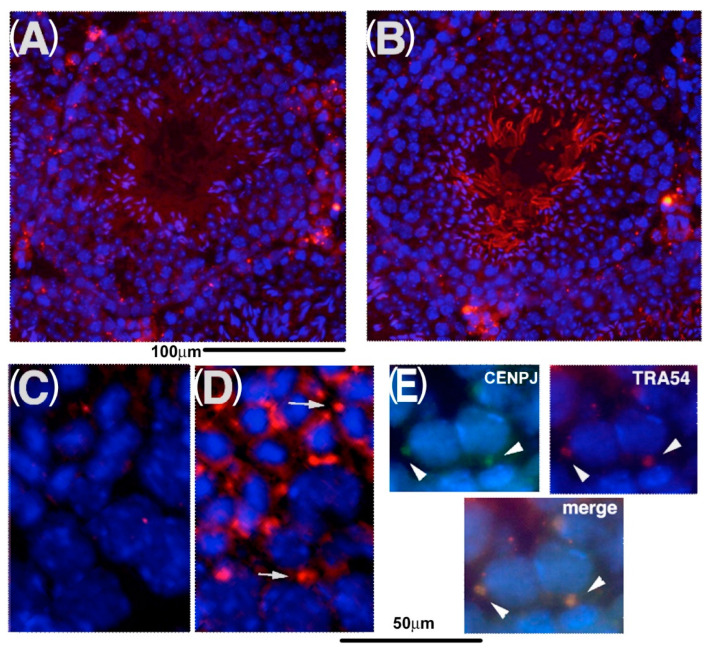
Localization of CENPJ in testicular cells. Sections of adult testes were stained with (**A**,**C**) pre-immune serum and (**B**) anti-CENPJ antibodies. Spermatocytes and round spermatids are shown in (**C**) and (**D**). White arrows in (**D**) indicate the signals of punctate patterns. (**E**) The signals of CENPJ are indicated with arrowheads and overlapped the signals of TRA54, which was stained in the chromatoid body [19]. Nuclei were stained with DAPI.

**Figure 8 ijms-23-09060-f008:**
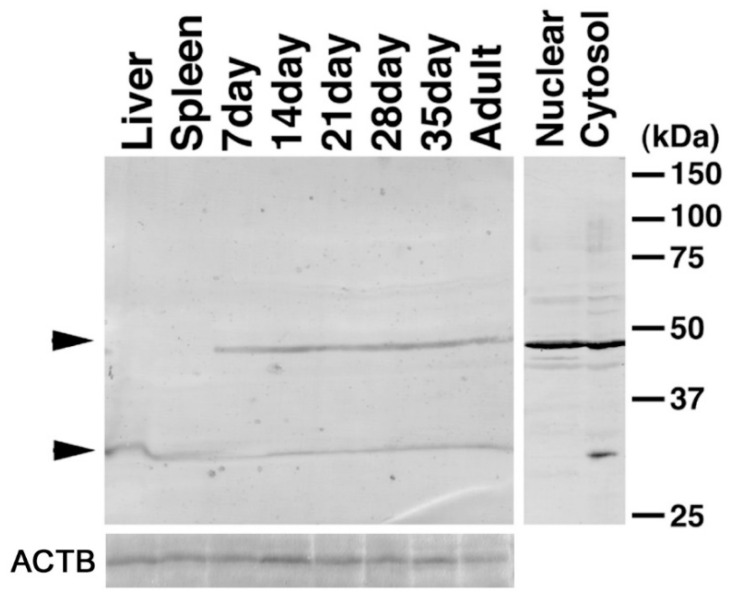
Western blotting of C1QBP in testes. The expression of C1QBP in testes was identified with the anti-C1QBP antibody using 50 μg lysates of liver, spleen, and testes of one-, two-, three-, four-, five-, and ten-week-old mice, and 30 μg lysates of nuclear and cytosol fractions. The arrowheads indicate signals of C1QBP. Molecular weights are shown on the right side.

**Figure 9 ijms-23-09060-f009:**
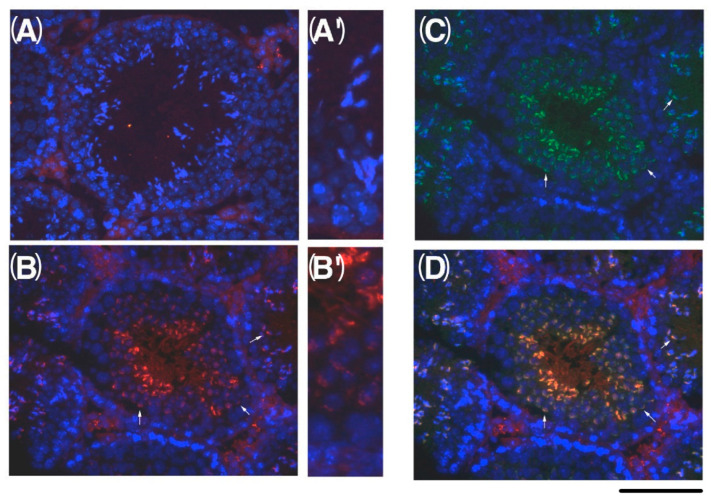
Localization of C1QBP in testicular cells. Sections of adult testes were stained with (**A**,**A’**) pre-immune serum, (**B**,**B’**) anti-C1QBP antibodies, and (**C**) FITC-PNA. Enlarged images of seminiferous tubules are shown in A’ and B’. (**D**) The merging of B and C shows C1QBP localized in acrosome. Nuclei were stained with DAPI. Bar indicates 100 μM.

**Figure 10 ijms-23-09060-f010:**
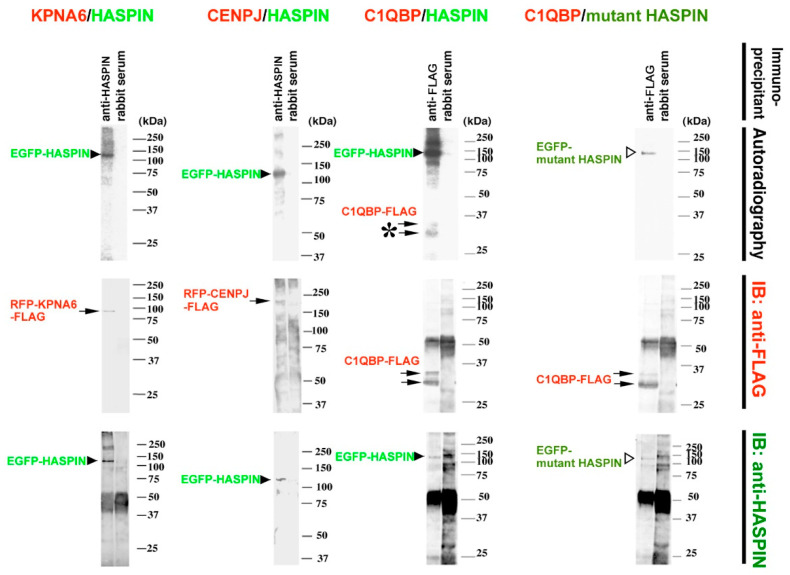
In vitro immune complex kinase assay. The recombinant *Egfp*-*Haspin* or mutant *Egfp-Haspin* expression vector was co-transfected in HEK 293 cells with *Rfp*- *Kpna6-Flag*, *Rfp*-*Cenpj-Flag*, or *C1qbp*-*Flag* expression vector, respectively. Immunoprecipitation was carried out using the anti-HASPIN anti-serum, anti-FLAG antibody, or normal rabbit serum, from the lysate of each culture cell. Genes expressed in HEK293 are shown above the panel. The antibody used for western blotting is indicated to the right of the panel. Arrowheads indicate EGFP-HASPIN. White arrows indicate EGFP-mutant HASPIN, other arrows indicate the signal of each recombinant protein fused with FLAG. Each immunoprecipitant contained EGFP-HASPIN, and each recombinant protein was expressed in HEK293s. The star indicates that FLAG-C1QBP was phosphorylated by GFP-HASPIN.

**Table 1 ijms-23-09060-t001:** The proteins binding to HASPIN identified by two hybrid system.

Registrated Gene Name (NCBI Accetion Number)	Domain in Cloned Gene	Number Of Clones Obtained
*CenpJ/CPAP* (NP_001014996)	Glycine repeat	5
*Kpna6/importin alpha 6* (NP_032494.3)	Whole	2
*Morn2* (NP_001347369)	MORN motif	2
*C1qbp/HABP1* (NP_031599.2)	Whole	1
*Ranbp9/RanBPM* (NP_001391577.1)	SPRY domain	1
*Tcp10/t-complex protein 10* (X58170)	Glycine repeat	1
*Aip* (NP_001263213.1)	Tetratricopeptide repeat	1
*Lims1* (NP_001180232.1)	LIM domain	1
*Pdlim5* (NP_001177781.1)	LIM domain	1
*Mad2l2* (NP_001292349.1)	HORMA domain	1

**Table 2 ijms-23-09060-t002:** Antibodies used in the current study.

Name	Reference
anti-KPNA6-rabbit IgG	this study
anti-CENPJ-rabbit IgG	this study
anti-C1QBP rabbit IgG	Muta et al. [69]
anti-HASPIN rabbit serum	Tanaka et al. [2]
Integrin β1 antibody	SantaCruz Biotechnology (SC-8978)
Anti-FLAG antibody	Sigma-Aldrich (F3165)
monoclonal antibody TRA98	Fukuda et al. [14,70]
monoclonal antibody TRA54	Pereira et al. [17]
rabibit anti-ACTB antibody	Cell Signaling (4967)
anti-rabbit IgG HRP-linked whole Ab (from donkey)	Amersham Biosciences (NA934V)
rabbit anti-rat IgG HRP-secondly antibody	DAKO (P0450)
donkey anti-rabbit IgG, biotinylated species specific antibody	Amersham Biosciences (SAB3700865)

## Data Availability

Data is contained within the article.

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
