# Peer review of "Analysis of Ser/Thr Kinase HASPIN-Interacting Proteins in the Spermatids"

_ijms, 2022, doi:10.3390/ijms23169060_

Round 1
Reviewer 1 Report
General comments
This work focuses on the binding partners of Haspin, a serine-threonine kinase predominantly expressed in spermatids but also present in meiotic and somatic cells. Haspin has been suggested to play distinct roles during interphase and cell division, on one hand regulating transcription (via a binary epigenetic switch or by modulating Pds5-cohesin associations with chromatin) and on the other hand facilitating recruitment of the Chromosome Passenger Complex (CPC) at the centromeric area of mitotic chromosomes.
The authors use a two-hybrid screen, to identify ten gene products that interact directly or indirectly with Haspin. Molecular interactions with CENPJ/CPAP, KPNA6/importin alpha 6, and C1QBP/HABP1 are confirmed by co-immunoprecipitation assays and the corresponding proteins localized in cells of the seminiferous tubule. Furthermore, it is shown that Haspin phosphorylates itself (which was previously known) and C1QBP/HABP1, a “protean” protein, which apparently plays various roles in different organelles.
The work is original and well executed. In addition, the literature has been exhaustively searched and the paper carefully written (not the norm nowadays). Some suggestions and clarifications that would improve the presentation and satisfy the sceptic are provided below.
Major points
1. The phosphorylation of C1QBP/HABP1 by Haspin is based on the data shown in Fig. 10. However, the evidence provided there is rather indirect: a doublet appearing in the 32P-autoradiogram when wild type Haspin is used (and not appearing with mutant Haspin) also appears upon immunoblotting with the anti-FLAG antibodies. The sceptic might ask for more convincing data here. Analysis by mass spectrometry would be the best method to make the point. Nevertheless, since phosphorylated peptides are often very labile and difficult to identify, a more practical approach along the same direction could be double immunoprecipitation (i.e., solubilizing the Haspin immunoprecipitate, re-precipitating the labeled protein with the anti-FLAG antibody and analyzing the products by autoradiography). It might be useful giving it a try, although the amounts of protein after two cycles of precipitation would be diminishing.
2. When compartment markers are used, the Haspin partners seem to partition in part with the nuclear and in part with the cytoplasmic fraction. However, the indirect immunofluorescence data leave no doubt that these proteins are localized in non-nuclear structures. An effort to localize unequivocally the kinase itself by good-old immunoelectron microscopy might not be a bad idea, provided that the antibodies are good.
Minor points
There several typos, some in critical points of the description (e.g., misspelling “FLAG”).
Author Response
To The Reviewer 1
Thank you for your careful review of our manuscript and for your thoughtful comments.
Major points
- The phosphorylation of C1QBP/HABP1 by Haspin is based on the data shown in Fig. 10. However, the evidence provided there is rather indirect: a doublet appearing in the 32P-autoradiogram when wild type Haspin is used (and not appearing with mutant Haspin) also appears upon immunoblotting with the anti-FLAG antibodies. The sceptic might ask for more convincing data here. Analysis by mass spectrometry would be the best method to make the point. Nevertheless, since phosphorylated peptides are often very labile and difficult to identify, a more practical approach along the same direction could be double immunoprecipitation (i.e., solubilizing the Haspin immunoprecipitate, re-precipitating the labeled protein with the anti-FLAG antibody and analyzing the products by autoradiography). It might be useful giving it a try, although the amounts of protein after two cycles of precipitation would be diminishing.
Answer: As the reviewer’s suggestion, we added the sentence Result and Discussion.
Results:
These results indicated that the kinase activity of HASPIN directly or indirectly phosphorylates C1QBP.
Discussion
It was suggested that C1QBP bound to HASPIN and was potentially a direct phosphorylation substrate of HASPIN. However, the other phosphorylation signals on high molecular weight observed in the kinase assay, suggested that other kinases might be involved in the phosphorylation of C1QBP.
- When compartment markers are used, the Haspin partners seem to partition in part with the nuclear and in part with the cytoplasmic fraction. However, the indirect immunofluorescence data leave no doubt that these proteins are localized in non-nuclear structures. An effort to localize unequivocally the kinase itself by good-old immunoelectron microscopy might not be a bad idea, provided that the antibodies are good.
Answer: We agree with the reviewer's suggestion.
We have tried many times to isolate good antibodies against mouse HASPIN. If successful, we think it will be useful for analysis of HASPIN.
Here, we would like to report novel molecules associated with HASPIN.
Minor points
There several typos, some in critical points of the description (e.g., misspelling “FLAG”).
Thank you for your suggestion. Our manuscript was submitted after English proofreading in MDIP. Here, we checked the words again throughout. It will be checked again by proofreading in English in MDIP if necessary according to the editor.
Reviewer 2 Report
In this paper, Maeda and colleagues analyzed three proteins that interacts with HASPIN, a kinase highly expressed in the testis: CENPJ/CPAP, KPNA6/importin alpha 6, and C1QBP/HABP1.
They found that these proteins are predominantly expressed in the meiotic and post-meiotic cells of mouse testis. Moreover, C1QBP is a substrate of HASPIN. As C1QBP is involved in centrosome formation, they concluded that HASPIN may be involved in the same pathway.
The focus of the paper is interesting; however, some revisions should be made:
- An editing should be made throughout the text, as many mistakes appear (different font and size, @ instead of µ, and so on;
- Why, in Fig.1, some picture lacks of the 18S bands?
- In Figs. 4, 6, and 8, the authors should add a Western blot also for a reference protein (as actin or GAPDH) as a loading control to normalize their data e add a graph on the protein expression levels;
- The authors assessed that “KPNA6 is also present in acrosomes in mature sperm”; however, it is difficult to confirm this with a conventional fluorescence microscope, so, the use a confocal microscope, or the execution of an in vitro acrosome reaction could be of helpful to ascertain this point;
- How can the authors explain the fact that they did not detect a signal for KPNA6 in spermatocytes or spermatogonia, but just in spermatids, while western blot showed a band also at 7 and 14 days, where spermatids are not supposed to be present? Moreover, in Fig. 5B’, some “non-spermatid” cells seem to be positive;
- As rodents’ spermatogenesis, as well as spermiogenesis, can be divided into different stages, an analysis on the localization of CENPJ, KPNA6, and C1QBP stage-specific should be added;
- In “Materials and Methods section” more details concerning the animals (including their housing condition, the ethical statement and the approval from the local ethical committee) should be added;
- In table 2, the reference codes for each antibody should be specified.
Author Response
We would like to thank the Editor and referees for their constructive comments, which have helped to improve the content of our paper. We have revised our manuscript accordingly.
To The Reviewer 2
- An editing should be made throughout the text, as many mistakes appear (different font and size, @ instead of µ, and so on;
Answer: Thank you for your suggestion. Our manuscript was submitted after English proofreading in MDIP. We checked the words again throughout. It will be checked again if necessary according to the editor.
- Why, in Fig.1, some picture lacks of the 18S bands?
Answer: Thank you for your suggestion. As you pointed out, we added positions of 28S 18S to the control.
- In Figs. 4, 6, and 8, the authors should add a Western blot also for a reference protein (as actin or GAPDH) as a loading control to normalize their data e add a graph on the protein expression levels;
Answer: Thank you for your suggestion. As you pointed out, we added normalize their data as a loading control to Figs. 4, and 8. Figures 4 and 6 were the same sample, so I wrote that in the figure legend of Fig. 6.
- The authors assessed that “KPNA6 is also present in acrosomes in mature sperm”; however, it is difficult to confirm this with a conventional fluorescence microscope, so, the use a confocal microscope, or the execution of an in vitro acrosome reaction could be of helpful to ascertain this point;
Answer: Thank you for your suggestion. As you pointed out, we added l figure S1. KPNA6 was not observed in sperm that have completed the acrosome reaction.
- How can the authors explain the fact that they did not detect a signal for KPNA6 in spermatocytes or spermatogonia, but just in spermatids, while western blot showed a band also at 7 and 14 days, where spermatids are not supposed to be present? Moreover, in Fig. 5B’, some “non-spermatid” cells seem to be positive;
Answer: Thank you for your suggestion. As you pointed out, we could find faint signals in in spermatocytes or spermatogonia, or in Sertoli cells or Leydig cells. We changed the paragraph as shown below.
No signal was observed in spermatocytes or spermatogonia, or in Sertoli cells or Leydig cells, when using our antibody to the N-terminus. KPNA6 was also present in sperm,
To
Faint signals were observed in spermatocytes or spermatogonia, or in Sertoli cells or Leydig cells, when using our antibody to the N-terminus. KPNA6 was also present in sperm,
- As rodents’ spermatogenesis, as well as spermiogenesis, can be divided into different stages, an analysis on the localization of CENPJ, KPNA6, and C1QBP stage-specific should be added;
Answer: I agree with the reviewer. We would like to report detailed analysis of each gene in the next papers.
- In “Materials and Methods section” more details concerning the animals (including their housing condition, the ethical statement and the approval from the local ethical committee) should be added;
Answer: Thank you for your suggestion. As you pointed out, we added the paragraph shown below in Materials and Methods.
4.1. Animals
C57BL/6J mice were purchased from Japan SLC (Shizuoka, Japan) and sacrificed by cervical dislocation immediately prior to experiments. All animal experiments conformed to the Guide for the Care and Use of Laboratory Animals and were approved by the Institutional Committee of Laboratory Animal Experimentation and Research Ethics Committee of Nagasaki International University (approval number 128). This article does not contain any experiments with human subjects performed by any of the authors. Mice were maintained under specific pathogen-free conditions in the animal experimentation facility at Nagasaki International University, with temperature and lighting controlled throughout the experimental period. Mice were provided with food and water ad libitum.
- In table 2, the reference codes for each antibody should be specified.
Answer: Thank you for your suggestion. As you pointed out, we added the information of antibodies in table2. We believe that the information presented here will lead you to the origins of the antibody used. Usage of antibodies were as shown in our manuscript.

Round 2
Reviewer 2 Report
The authors responded to almost all the comments, ameliorating the quality of the MS. It is now acceptable for publication in IJMS.